# Effectiveness of Early Radical Cystectomy for High-Risk Non-Muscle Invasive Bladder Cancer

**DOI:** 10.3390/cancers14153797

**Published:** 2022-08-04

**Authors:** Elliott Diamant, Mathieu Roumiguié, Alexandre Ingels, Jérôme Parra, Dimitri Vordos, Anne-Sophie Bajeot, Emmanuel Chartier-Kastler, Michel Soulié, Alexandre de la Taille, Morgan Rouprêt, Thomas Seisen

**Affiliations:** 1Sorbonne Université, Department of Urology, GRC n°5 Predictive Onco-Urology, AP-HP, Pitié-Salpêtrière Hospital, 75013 Paris, France; jerome.parra2@gmail.com (J.P.); emmanuel.chartier-kastler@aphp.fr (E.C.-K.); mroupret@gmail.com (M.R.); thomas.seisen@gmail.com (T.S.); 2Department of Urology, CHU-Institut Universitaire du Cancer-Oncopôle, 31000 Toulouse, France; roumiguie_mathieu@yahoo.fr (M.R.); anne-sophie_bajeot@hotmail.fr (A.-S.B.); soulie.m@chu-toulouse.fr (M.S.); 3Department of Urology, University Hospital Henri Mondor, APHP, UPEC, 94000 Créteil, France; alexandre.ingels@aphp.fr (A.I.); dvordos@hotmail.com (D.V.); adelataille@hotmail.com (A.d.l.T.)

**Keywords:** urothelial carcinoma, non-muscle invasive bladder cancer, cystectomy, complication, upstaging, survival analysis

## Abstract

**Simple Summary:**

There is currently contradictory evidence available regarding the use of early radical cystectomy for high-risk non-muscle-invasive bladder cancer that can be performed either upfront or in a delayed setting after BCG failure. Thus, we aimed to compare the perioperative and oncological outcomes between patients who underwent upfront vs. delayed early radical cystectomy. Our results suggest similar perioperative outcomes between the two treatment groups, with an increased risk of pathological upstaging after upfront early radical cystectomy that did not impact survival, as compared to delayed early radical cystectomy. However, further studies are needed to determine whether a subgroup of patients might still benefit from upfront early radical cystectomy, given the highly heterogeneous prognosis of this population. This supports the initial use of intravesical instillations of BCG for patients with high-risk non-muscle-invasive bladder cancer, but further studies are needed to determine if any specific subgroup could still benefit from upfront early radical cystectomy.

**Abstract:**

Purpose: The purpose of this study is to compare perioperative and oncological outcomes of upfront vs. delayed early radical cystectomy (eRC) for high-risk non-muscle-invasive bladder cancer (HR-NMIBC). Methods: All consecutive HR-NMIBC patients who underwent eRC between 2001 and 2020 were retrospectively included and divided into upfront and delayed groups, according to the receipt or not of BCG. Perioperative outcomes were evaluated and the impact of upfront vs. delayed eRC on pathological upstaging, defined as ≥pT2N0 disease at final pathology, was assessed using multivariable logistic regression. Recurrence-free (RFS), cancer-specific (CSS) and overall survival (OS) were compared between upfront and delayed eRC groups using inverse probability of treatment weighting (IPTW)-adjusted Cox model. Results: Overall, 184 patients received either upfront (*n* = 87; 47%) or delayed (*n* = 97; 53%) eRC. No difference was observed in perioperative outcomes between the two treatment groups (all *p* > 0.05). Pathological upstaging occurred in 55 (30%) patients and upfront eRC was an independent predictor (HR = 2.65; 95% CI = (1.23–5.67); *p* = 0.012). In the IPTW-adjusted Cox analysis, there was no significant difference between upfront and delayed eRC in terms of RFS (HR = 1.31; 95% CI = (0.72–2.39); *p* = 0.38), CSS (HR = 1.09; 95% CI = (0.51–2.34); *p* = 0.82) and OS (HR = 1.19; 95% CI = (0.62–2.78); *p* = 0.60). Conclusion: our results suggest similar perioperative outcomes between upfront and delayed eRC, with an increased risk of upstaging after upfront eRC that did impact survival, as compared to delayed eRC.

## 1. Introduction

Bladder cancer is the 10th most frequent malignancy worldwide, with 549,000 new cases diagnosed each year [1]. Although some patients present with muscle-invasive bladder cancer (MIBC), those with non-muscle invasive bladder cancer (NMIBC) at initial diagnosis still represent 75% of the disease burden [2]. From a pathological perspective, NMIBC remains confined to the mucosa and lamina propria without invading the bladder muscle, but prognosis is highly heterogeneous, depending on many baseline characteristics. Several predictive models have been developed to risk-stratify NMIBC patients with regard to the probability of recurrence and progression after initial transurethral resection of the bladder tumor (TURBT) [3]. Interestingly, those with high-risk NMIBC (HR-NMIBC) account for approximately 20%, encompassing individuals with pT1 or high-grade disease, carcinoma in situ (CIS) and other aggressive features, such as multifocality, large tumor size, histologic variants or lymphovascular invasion (LVI).

After initial TURBT, followed by a second look procedure only for T1 high-grade tumors, the gold-standard treatment for HR-NMIBC consists of the delivery of intravesical immunotherapy using induction and maintenance course of Bacillus Calmette et Guerin (BCG) [2]. Although the adequate use of BCG could reduce the risk of progression down to approximately 6% after 5 years in HR-NMIBC patients, it has been reported to still be up to almost 15% in those with combined multiple adverse pathological features [4]. Thus, based on retrospective evidence, early radical cystectomy (eRC), meaning radical cystectomy for patients that present with NMIBC, has been proposed to provide better disease control and prolong survival in selected cases of HR-NMIBC, where oncological benefits outweigh perioperative risks related to this procedure [5,6,7,8]. Recently, the BRAVO feasibility study showed that conducting a randomized controlled trial comparing eRC to intravesical BCG is likely to be challenging but secondary outcomes confirmed the role of eRC for some HR-NMIBC patients by revealing a 10% risk of harboring lethal disease at final pathology [9].

However, there is currently contradictory evidence available regarding the use of eRC that can be performed either upfront or in a delayed setting after BCG failure. Indeed, some reports suggested a survival benefit with the use of upfront [5,6,7,8], or delayed eRC [10,11], while others found no difference between either of them [12,13]. In addition, very few data have been published on the perioperative results of upfront or delayed eRC [13,14,15]. Against this backdrop, our aim was to evaluate eRC by comparing perioperative and oncological outcomes obtained after upfront or delayed procedures for HR-NMIBC.

## 2. Materials and Methods

### 2.1. Population Study

All patients older than 18 years old who underwent eRC for HR-NMIBC with or without receiving intravesical instillation of BCG before surgery at three tertiary-care centers in France between January 2001 and May 2020 were retrospectively included in this study. The diagnosis of HR-NMIBC was carried out using TURBT specimens that showed high-grade disease classified as either pTis, pTa or pT1. Only those with intradiverticular HR-NMIBC were excluded, given the risk of underestimating the disease stage at initial diagnosis.

### 2.2. Technical Aspects of eRC

Using the Da Vinci Si or Xi platform, eRC was performed through either an open, laparoscopic or robot-assisted approach and consisted of cystoprostatectomy for men and anterior pelvectomy for women. A concomitant urethrectomy was carried out only for prostatic urethra or bladder neck involvement in men and women, respectively. Pelvic lymph node dissection up to the crossing of the ureter with the primary iliac artery was left to the surgeon’s discretion. Ileal conduit, orthotopic or heterotopic enterocystoplasty and simple skin ureterostomies were used for urinary diversion, depending on patient and tumor characteristics.

### 2.3. Treatment Groups

Patients who underwent eRC without receiving any intravesical instillation of BCG before surgery were included in the upfront eRC group, while those who received at least an induction course with six intravesical instillations of BCG before surgery were included in the delayed eRC group.

### 2.4. Study Endpoints

Perioperative outcomes of patients who received upfront vs. delayed eRC were evaluated using intraoperative and postoperative data, while oncological outcomes were compared between the two treatment groups using pathological and survival data.

### 2.5. Data Collection

After institutional review board approval, all information available from HR-NMIBC diagnosis to the last follow-up visit or death were collected at the three participating centers using a computerized database.

Baseline characteristics included age, gender, body mass index (BMI), ASA score, smoking status, information from initial TURBT, such as cT stage, tumor focality and size, presence of concomitant CIS or lymphovascular invasion (LVI), performance of 2nd look TURBT, number of intravesical instillations of BCG before delayed eRC, as well as time from initial TURBT to eRC.

With regard to perioperative outcomes assessment, intraoperative data included surgical approach, performance of concomitant urethrectomy or lymph node dissection, urinary diversion type, median operative time and blood loss volume with delivery of blood transfusion during eRC, while post-operative data included median length of stay with hemoglobin and creatinine levels at discharge, in-hospital delivery of blood transfusion, and the occurrence of any adverse events within 30 days after eRC, graded according to the Clavien–Dindo classification.

With regard to oncological outcome assessment, pathological data included pT and pN stage, as well as surgical margins status and the presence of concomitant CIS or LVI at final pathology, while survival data included recurrence free survival (RFS), cancer-specific survival (CSS) and overall survival (OS), defined as the time from eRC to the event of interest. It is noteworthy that surveillance regimens were performed in accordance with the principles established by the latest EAU guidelines [2]. Importantly, recurrence was defined as tumor relapse in the operative field, pelvic lymph nodes and/or distant metastatic sites. With regard to mortality, the cause of death was determined by the referent attending physician based on the medical chart and death certificate review. Only deaths from bladder cancer were coded as cancer-specific events and all patients who died from bladder cancer had prior disease recurrence.

### 2.6. Statistical Analyses

Medians with interquartile ranges (IQRs), as well as numbers with proportions, were used to report the continuous and categorical variables, respectively. Comparisons of these continuous and categorical variables between the patients treated with upfront or delayed eRC were conducted using Student’s and Chi2 tests, respectively. Uni- and multivariable logistic regression analyses with the purposeful selection method were also performed to determine the odds ratio (OR) and its 95% confidence interval (95% CI) of the impact of upfront or delayed eRC on the risk of pathological upstaging, defined as ≥pT2N0 disease using the eRC specimen [16].

With regard to survival analyses, the Kaplan–Meier method was first used to present RFS, CSS and OS associated with eRC overall, regardless of whether the procedure was performed in an upfront or delayed setting. Observed differences in the baseline characteristics between patients who underwent upfront vs. delayed eRC were further controlled for the inverse probability of treatment weighting (IPTW)-adjusted analyses. Specifically, propensity scores obtained from a multivariable logistic regression model that predicted the probability of undergoing upfront vs. delayed eRC were used to weigh each patient, with the aim of balancing out the baseline characteristics between the two treatment groups [17]. The method of Hosmer and Lemeshow was used to ensure model adjustment [18]. The distribution of covariates between the two treatment groups before and after IPTW adjustment was estimated by calculating standardized differences [19]. A standardized difference of <10% indicated no significant difference between the patients who underwent upfront or delayed eRC. In addition, kernel density plots were used to compare propensity score distribution between the two treatment groups. Finally, IPTW-adjusted Kaplan–Meier curves were computed to compare RFS, CSS and OS between patients who received upfront vs. delayed eRC [20]. To test for RFS, CSS and OS equality between the two treatment groups, we used IPTW-adjusted log-rank tests. In addition, post-weighting univariable Cox proportional hazards regression analyses were conducted to calculate ITPW-adjusted hazard ratios (HRs) with corresponding 95% CI for RFS, CSS and OS.

All statistical analyses were performed using Stata software (Stata/SE version 15.1, Stata Corp LLC, Lakeway Drive College Station, TX, USA). Significance was defined as *p* < 0.05.

## 3. Results

Overall, 184 patients who underwent upfront (*n* = 87; 47%) or delayed (*n* = 97; 53%) eRC for HR-NMIBC were included in this study.

### 3.1. Baseline Characteristics

Baseline characteristics of the included patients, stratified according to the receipt of upfront or delayed eRC, are reported in Table 1. With regard to patient characteristics, median age was significantly higher in the group of those who underwent upfront eRC, as compared to that in the delayed eRC group (68 vs. 64 years, *p* = 0.024). The male to female ratio was 19:1 (175 men vs. 9 women), with no significant difference in gender distribution between the two treatment groups (*p* = 0.610). In addition, there was no significant difference in median BMI (*p* = 0.945), ASA score (*p* = 0.063) or smoking status (*p* = 0.817) between the patients who underwent upfront or delayed eRC.

With regard to disease characteristics, all the included patients had high-grade NMIBC at TURBT and there was no significant difference in cT stage (*p* = 0.076), concomitant CIS (*p* = 0.388) or LVI (*p* = 0.077) between the two treatment groups. However, patients treated with upfront eRC were more likely to have multifocal (57% vs. 37%; *p* = 0.001) and ≥3 cm (28% vs. 7%; *p* = 0.001) tumors, while those treated with delayed eRC were more likely to undergo second look TURBT (92% vs. 76%; *p* < 0.001).

### 3.2. Perioperative Outcomes

#### 3.2.1. Intraoperative Outcomes

Intraoperative outcomes of the included patients, stratified according to the receipt of upfront or delayed eRC, are reported in Table 2. No significant difference was observed between the two treatment groups (all *p* > 0.05). Overall, 77% of the patients underwent open RC with lymph node dissection but without urethrectomy in 96% and 91% of the cases, respectively. The main urinary diversion types were ileal conduit and enterocystoplasty in 51% and 44% of the cases, respectively. The median operative time was 300 (258–390) minutes, with a median blood loss of 800 (400–1000) mL and a risk of intraoperative transfusion of 23%.

#### 3.2.2. Postoperative Outcomes

Postoperative outcomes of the included patients, stratified according to the receipt of upfront or delayed eRC, are reported in Table 3. No significant difference was observed between the two treatment groups (all *p* > 0.05). Overall, the median length of stay was 17 (14–21) days. The risk of postoperative transfusion was 23%, with a median hemoglobin and creatinine at discharge of 11 (10–13) g/dL and 93 (74–113) ng/mL, respectively. The risk of early postoperative complications was 60%, of which 35% were grade 1–2 complications, according to the Clavien–Dindo classification. However, 23% of the patients had a grade 3 or 4 complication that required an intensive care unit transfer or a surgical management for evisceration and digestive or urinary fistula.

### 3.3. Oncological Outcomes

#### 3.3.1. Pathological Outcomes

Cross-tabulation of cT stage from TURBT specimen and pT stage from eRC is reported in Table 4. The concordances between cTa/pTa, cTis/pTis and cT1/pT1 stages were 31.5%, 56% and 24%, respectively. A total of 55 (30%) patients had pathological upstaging, including 10% (*n* = 18) of pT2N0 disease and 20% (*n* = 37) of ≥pT3N0 disease. The vast majority of these patients had cT1 disease at TURBT (*n* = 46; 84%) but five (10%) and four (6%) patients had cTa and cTis disease at TURBT, respectively.

Other pathological outcomes of the included patients, stratified according to the receipt of upfront or delayed eRC, are reported in Table 5. Patients who underwent upfront eRC had a higher risk of pathological upstaging (35% vs. 23%; *p* = 0.046), whereas those who had delayed eRC had a higher probability of having pT0 disease (26% vs. 10%; *p* = 0.046). However, there was no significant difference between the two treatment groups in terms of pelvic lymph node invasion (8% vs. 6%; *p* = 0.757), concomitant CIS (46% vs. 42%; *p* = 0.613), LVI (6% vs. 1%; *p* = 0.057) or positive surgical margin (3% vs. 7%; *p* = 0.260).

Results from the univariable and multivariable analyses of the predictors of pathological upstaging are presented in Table 6. In the multivariable analysis with the purposeful selection method, patients treated with upfront eRC had a significantly increased risk of pathological upstaging, as compared to those treated with delayed eRC (OR = 2.65; 95% CI = (1.23–5.67); *p* = 0.012). Other independent predictors of pathological upstaging included BMI (OR = 1.29; 95% CI = (1.22–1.97); *p* = 0.008), history of cigarette smoking (OR = 5.40; 95% CI = (1.45–20.1); *p* = 0.012), and concomitant CIS (OR = 1.46; 95% CI = (1.22–1.95); *p* = 0.037).

#### 3.3.2. Survival Outcomes

The median follow-up in the overall cohort was 65 (36–127) months. Following eRC, 5-year RFS (Figure 1), CSS (Figure 2) and OS (Figure 3) rates were 63.5% (95% CI = (53.9–71.5)), 75.9% (95% CI = (66.8–82.8)) and 68.4% (95% CI = (59.3–75.9)), respectively. With regard to recurrences, 50 patients (27.2%) relapsed within a median time of 11 months (IQR, 7–20.8) after eRC. The majority of recurrences were extra-urothelium (*n* = 32; 64%), with lymph node, lung, liver, peritoneal, and/or bone metastases. Intra-urothelium recurrences in the upper urinary tract or urethra were observed in 18 patients (36%). Systemic chemotherapy was used in 20 patients (40%), while local treatment using either surgery or radiotherapy was performed in 24 patients (48%). Exclusive palliative care was delivered to six patients (12%).

Results from the propensity score model that predicted the receipt of upfront vs. delayed eRC are reported in Table 7 (*p* = 0.499 using the Hosmer and Lemeshow method). Following IPTW adjustment, standardized differences between the patients treated with upfront or delayed eRC were less than 10% for all covariates, except for the use of 2nd look TURB (15.4%, Figure 4). In addition, Kernel density plots showed that the propensity score distribution achieved adequate balance between the two treatment groups after IPTW adjustment (Figure 5). This indicated that patients who underwent upfront or relayed eRC were subsequently comparable.

With regard to RFS, the IPTW-adjusted Kaplan–Meier curves (Figure 6) showed that 5-year RFS was 55.7% and 66.7% after upfront and delayed eRC (*p* = 0.38 by IPTW-adjusted log rank test), respectively. In the IPTW-adjusted Cox regression analysis, there was no significant difference in RFS between the patients who received upfront or delayed eRC (HR = 1.31; 95% CI = (0.72–2.39); *p* = 0.38).

With regard to CSS, the IPTW-adjusted Kaplan–Meier curves (Figure 7) showed that 5-year CSS was 75.1% and 72.9% after upfront and delayed eRC (*p* = 0.82 by IPTW-adjusted log rank test), respectively. In the IPTW-adjusted Cox regression analysis, there was no significant difference in CSS between the patients who received upfront or delayed (HR = 1.09; 95% CI = (0.51–2.34); *p* = 0.82).

With regard to OS, the IPTW-adjusted Kaplan–Meier curves (Figure 8) showed that 5-year CSS was 67.0% and 68.4% after upfront and delayed eRC (*p* = 0.60 by IPTW-adjusted log rank test), respectively. In the IPTW-adjusted Cox regression analysis, there was no significant difference in OS between the patients who received upfront or delayed (HR = 1.19; 95% CI = (0.62–2.78); *p* = 0.60).

## 4. Discussion

Patients diagnosed with HR-NMIBC have a highly heterogenous prognosis, depending on the number of adverse pathological features observed at TURBT. Thus, optimizing the management of these patients is of utmost importance. Against this backdrop, we evaluated the relevance of performing eRC either in the upfront or delayed setting. Overall, we observed similar perioperative outcomes between the two treatment groups. However, with regard to oncological outcomes, there was a more than 2.5-fold increase in the risk of pathological upstaging with the use of upfront eRC, although it did not impact survival with similar RFS, CSS and OS, as compared to delayed eRC. These results suggest that HR-NMIBC patients can safely avoid undergoing upfront eRC to receive first-line intravesical BCG and delayed eRC for refractory disease.

It is noteworthy that our study is the first report to evaluate the intraoperative outcomes of eRC. We observed that eRC was mostly performed using an open approach, with a concomitant pelvic lymph node dissection and balanced distribution between ileal conduit and enterocystoplasty. Median operative time and intraoperative blood loss were consistent with that observed for RC performed for MIBC [21]. However, with regard to postoperative outcomes, the risk of early postoperative complications after eRC has been previously reported in three retrospective studies [13,14,15]. Interestingly, ≥ grade 3 complications occurred in 6% to 20% of the included patients [14], with no significant difference the between patients treated with upfront or delayed eRC in the study by Ali-El-Dein et al. [13]. Although our report suggests a higher risk of ≥ grade 3 complications up to 25%, we accordingly did not find any significant difference between the two treatment groups. Median length of stay ranged from 7 to 9 days in the literature [14], and was, therefore, lower than 15 days observed in our study, but this could be explained by the differences in the postoperative discharge policy in health systems worldwide. Importantly, no information was available on the readmission rate in our database, but Tully et al. found up to 9% readmission within 30 days of discharge in the National Cancer Database [15].

With regard to pathological upstaging, we observed that one third of HR-NMIBC patients harbored MIBC at eRC, of which more than half had extravesical disease and/or pelvic lymph node invasion. These results are in line with most of the literature, showing pathological upstaging rates ranging from 20% to 50% [5,6,12,22]. However, the BRAVO randomized controlled trial suggested that only 10% of patients had MIBC at eRC, despite being limited by a small sample size [9]. In addition, Hautmann et al. reported that approximately 10% of HR-NMIBC patients had ≥pT3 disease at final pathology in a series of 274 individuals who underwent eRC [7]. Interestingly, our study is the first to show a greater risk of pathological upstaging in patients who underwent upfront vs. delayed eRC in a multivariable analysis. Indeed, some reports suggested either decreased [7], or similar risk, of pathological upstaging with the use of upfront eRC as compared to delayed eRC [23], but a recent systematic review showed that the timing of eRC had no significant impact overall [24]. Although the increased risk of pathological upstaging after upfront eRC was independent of the use of second look TURBT in our study, patients who underwent delayed eRC were more likely to receive it, with a potential direct impact on these results. Indeed, a recent meta-analysis showed that second look TURBT could evidence residual tumors in up to 50% of HR-NMIBC patients, with an upstaging rate of 10% [25]. Other endoscopic techniques, such as “en bloc” resection of the tumor [26,27,28], or hexaminolevulinate lumino-fluorescence cystoscopy, could also improve disease staging [29,30,31,32]. However, no information on the use of these techniques was available in our database. Regarding the development of other disease staging methods, bladder MRI, with the use of the standardized Vesical Imaging Reporting and Data System (VIRADS) score, has been evaluated in a meta-analysis of 6 studies and 1770 patients, showing a sensitivity of 83% and a specificity of 90% for the detection of muscle invasion [33]. Finally, as opposed to pathological upstaging, the probability of having pT0N0 disease at final pathology was 17% in our study, which was also in line with the current literature, showing rates ranging from 0% to 18% [5,34,35,36].

Overall, we observed similar oncological outcomes than that previously reported with the use of eRC [35,37,38,39,40,41,42], but more contradictory findings have been published on the comparison of upfront vs. delayed setting. Although three studies including 143 to 204 patients did not also find any significant difference in CSS and OS between the two treatment groups [11,12,13], other reports have suggested a benefit with the use of either upfront or delayed eRC. Specifically, only 2 reports including more than 120 patients showed prolonged CSS or OS after delayed eRC [6,10], but 5 reports including 105 to 4900 patients showed prolonged RFS, CSS and/or OS after upfront eRC [5,7,8,15,34].

The results of our study must be interpreted within the limitations of its retrospective design. Although it was a large multicenter analysis, including almost 200 patients that were operated on in 3 high-volume three tertiary-care centers over a period of 20 years, some information was not available in our database. In particular, indications for upfront eRC remain unknown and since it is not a standard first-line treatment, it is conceivable that these patients presented with more aggressive pathological features at initial diagnosis than those treated with intravesical instillations of BCG, followed by delayed eRC. For example, there were missing data on tumor focality or size for 25% each of the included patients, while the presence of LVI remained unknown for 40% of the patients. Our results may, therefore, be subject to selection bias, disadvantaging the upfront eRC arm. In addition, although all patients in the delayed eRC group received an induction course of BCG, the exact number of intravesical instillations during the maintenance treatment remains unknown. Moreover, despite a median follow-up of more than 5 years, median RFS, CSS, and OS were not reached and a longer follow-up would provide more accurate conclusions on the oncological outcomes of upfront or delayed eRC. Nevertheless, the results of our study allow for hypothesizing that upfront eRC may not lead to any survival benefit in HR-NMIBC patients, as compared to those undergoing initial intravesical instillations of BCG with or without delayed eRC, given that, despite the fact that individuals who did not require delayed eRC were not included, they are likely to have an even better prognosis. Finally, we focused our report on the perioperative and oncological outcomes but quality of life is also highly relevant, with expected differences favoring delayed eRC that allows for longer preservation of the bladder.

## 5. Conclusions

Overall, our results suggest similar perioperative outcomes for upfront and delayed eRC in HR-NMIBC patients. Although there was an increased risk of pathological upstaging after upfront eRC, we did not observe any significant difference in RFS, CSS or OS, as compared to delayed eRC. This supports the initial use of intravesical instillations of BCG for HR-NMIBC patients, but further studies are needed to determine if any specific subgroup could still benefit from upfront eRC, given the highly heterogeneous prognosis of this population. The enhancement of invasive and non-invasive diagnostic tools could help to improve HR-NMIBC patient selection, especially in the context of the development of many conservative treatments for BCG-refractory disease.

## Figures and Tables

**Figure 1 cancers-14-03797-f001:**
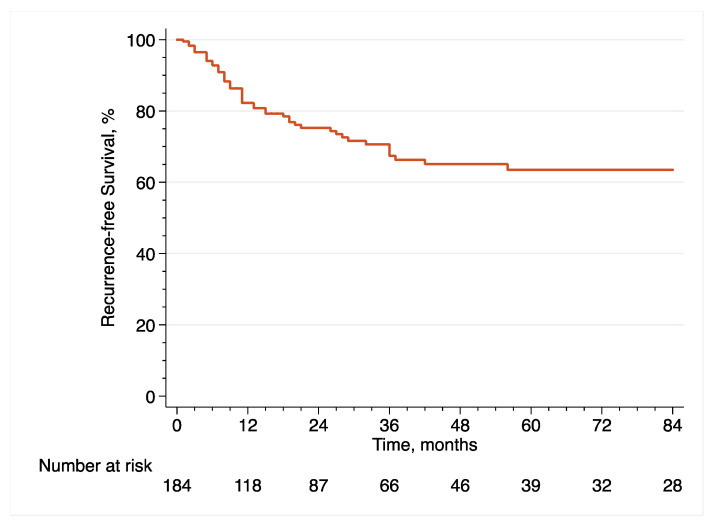
Kaplan–Meier curve that analyses RFS of included patients treated with eRC for HR-NMIBC.

**Figure 2 cancers-14-03797-f002:**
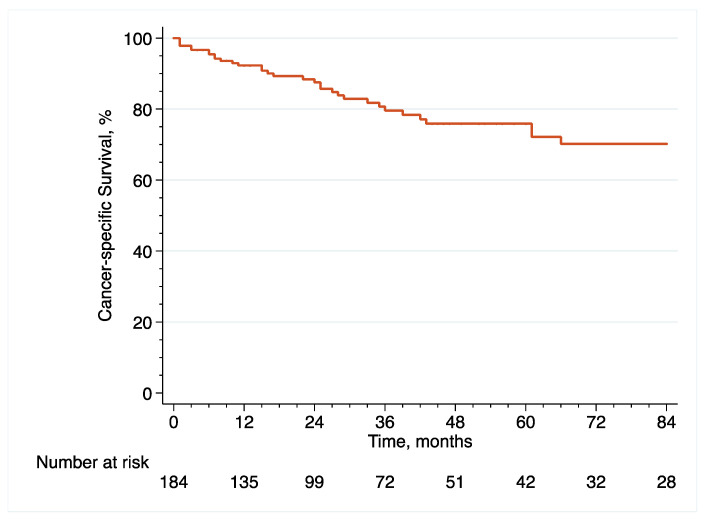
Kaplan–Meier curve that analyses CSS of included patients treated with eRC for HR-NMIBC.

**Figure 3 cancers-14-03797-f003:**
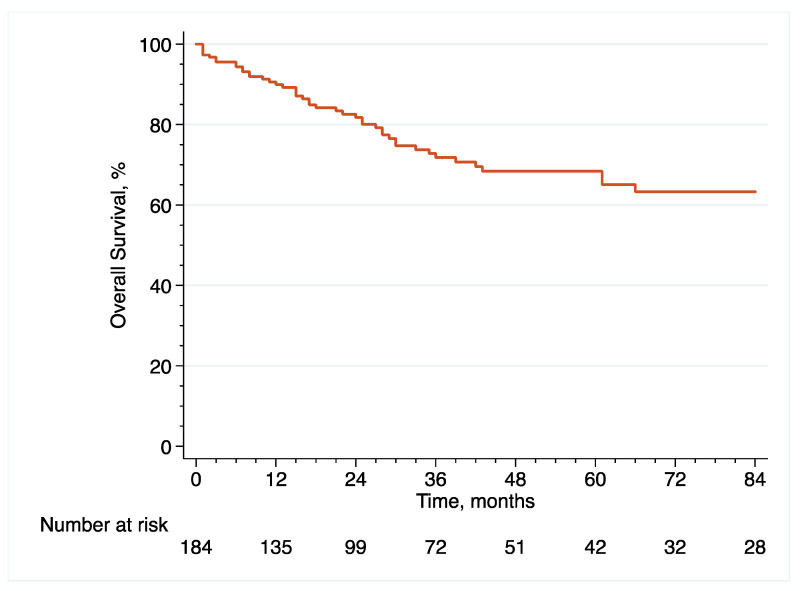
Kaplan–Meier curve that analyses OS of included patients treated with eRC for HR-NMIBC.

**Figure 4 cancers-14-03797-f004:**
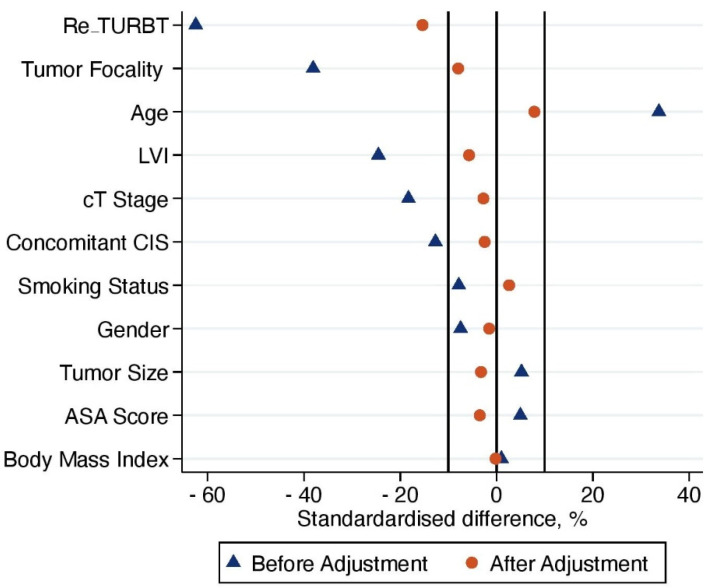
Impact inverse probability of treatment weighting (IPTW) adjustment on the distribution of baseline characteristics of included patients treated with upfront or delayed eRC for HR-NMIBC.

**Figure 5 cancers-14-03797-f005:**
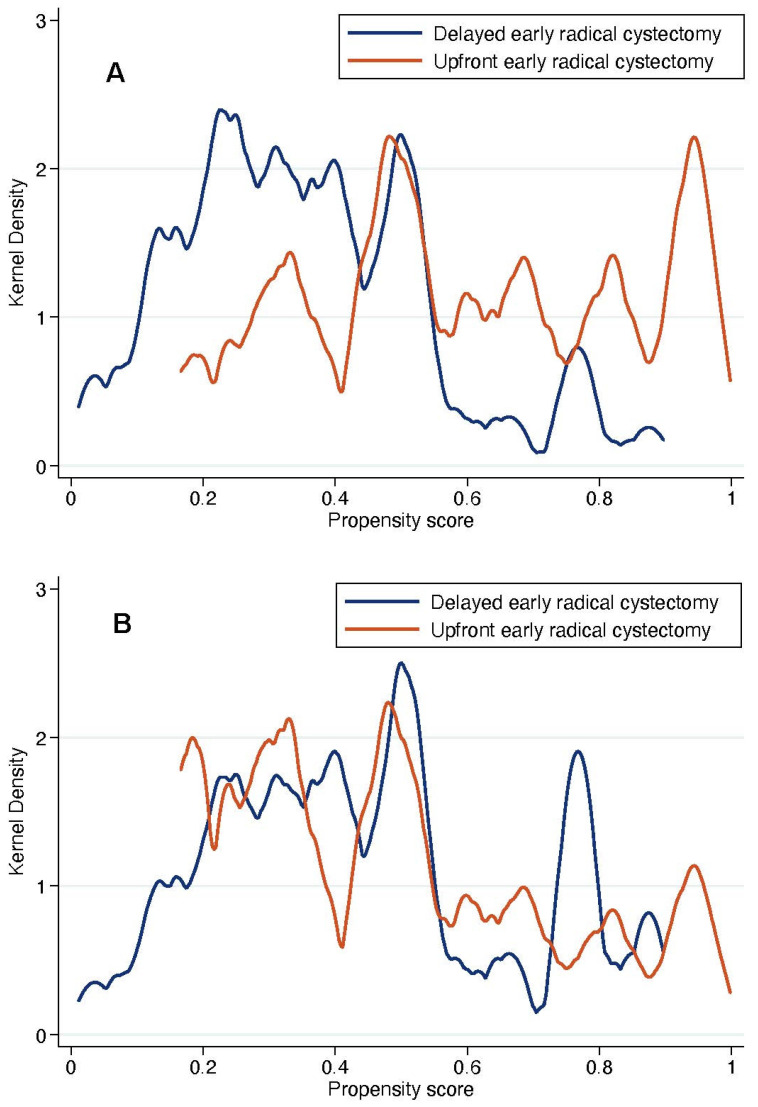
Kernel density plots that show the distribution of propensity scores in included patients treated with upfront or delayed eRC groups before (**A**) and after (**B**) adjustment by inverse probability of treatment weighting (IPTW) method.

**Figure 6 cancers-14-03797-f006:**
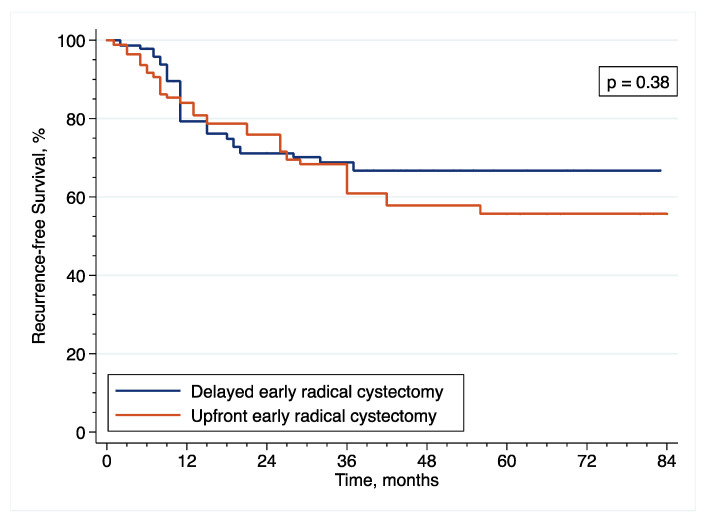
Inverse probability of treatment weighting (IPTW)-adjusted Kaplan–Meier curves that compare the RFS of included patients treated with upfront or delayed eRC for HR-NMIBC.

**Figure 7 cancers-14-03797-f007:**
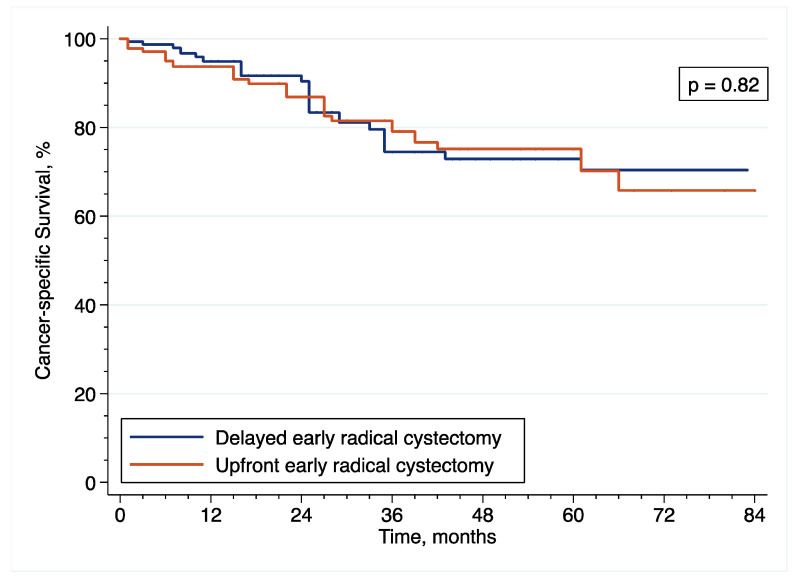
Inverse probability of treatment weighting (IPTW)-adjusted Kaplan–Meier curves that compare the CSS of included patients treated with upfront or delayed eRC for HR-NMIBC.

**Figure 8 cancers-14-03797-f008:**
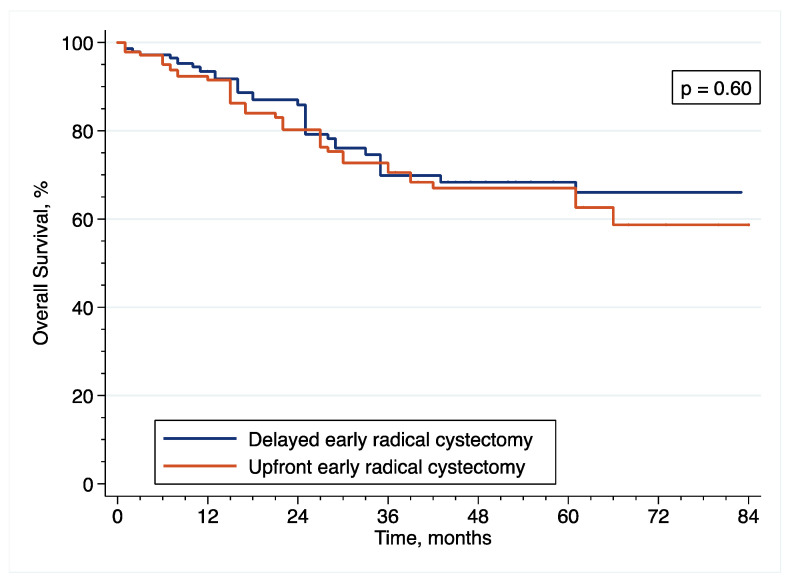
Inverse probability of treatment weighting (IPTW)-adjusted Kaplan–Meier curves that compare the OS of included patients treated with upfront or delayed eRC for HR-NMIBC.

**Table 1 cancers-14-03797-t001:** Baseline characteristics of included patients treated with upfront or delayed eRC for HR-NMIBC. Bold: It is a significant result.

Preoperative Characteristics	Total	Upfront eRC	Delayed eRC	*p*
	*(N = 184; 100%)*	*(N = 87; 47.3%)*	*(N = 97; 52.7%)*	
**Median age, years (IQR)**	66 (58–72)	68 (60–73)	64 (57–70)	**0.024**
**Gender, *n* (%)**				0.610
**Male**	175 (95)	82 (94)	93 (96)	
**Female**	9 (5)	5 (6)	4 (4)	
**Median BMI, kg/m^2^ (IQR)**	25 (23–29)	25 (24–29)	25 (23–29)	0.945
**ASA score, *n* (%)**				0.063
**≤2**	126 (68)	61 (70)	65 (67)	
**≥3**	51 (28)	20 (23)	31 (32)	
**NA**	7 (4)	6 (7)	1 (1)	
**Smoking status, *n* (%)**				0.817
**Never**	43 (24)	22 (25)	21 (22)	
**Former**	19 (10)	8 (9)	11 (11)	
**Current**	116 (63)	55 (63)	61 (63)	
**NA**	6 (3)	2 (3)	4 (4)	
**cT stage, *n* (%)**				0.076
**cTa**	48 (26)	23 (26)	25 (26)	
**cTis**	18 (10)	4 (5)	14 (14)	
**cT1**	118 (64)	60 (69)	58 (60)	
**Tumor focality, *n* (%)**				**0.001**
**Unifocal**	51 (28)	26 (29)	25 (26)	
**Multifocal**	85 (46)	49 (57)	36 (37)	
**NA**	48 (26)	12 (14)	36 (37)	
**Tumor size, *n* (%)**				**0.001**
**<3 cm**	103 (56)	43 (49)	60 (62)	
**≥3 cm**	31 (17)	24 (28)	7 (7)	
**NA**	50 (27)	20 (23)	30 (31)	
**Concomitant CIS, *n* (%)**				0.388
**Absent**	89 (48)	45 (52)	44 (45)	
**Present**	95 (52)	42 (48)	53 (55)	
**LVI, *n* (%)**				0.077
**Absent**	101 (55)	52 (60)	49 (51)	
**Present**	10 (5)	7 (8)	3 (3)	
**NA**	73 (40)	28 (32)	45 (46)	
**Second look TURB, *n* (%)**				**<0.001**
**Yes**	21 (12)	19 (22)	2 (2)	
**No**	155 (84)	66 (76)	89 (92)	
**NA**	8 (4)	2 (2)	6 (6)	
**Median interval between initial diagnosis and eRC, months (IQR)**	11 (5–31)	5 (3–14)	19 (10–47)	**<0.001**

**Table 2 cancers-14-03797-t002:** Intraoperative outcomes of included patients treated with upfront or delayed eRC for HR-NMIBC.

Intraoperative Outcomes	Total	Upfront eRC	Delayed eRC	*p*
	*(N = 184; 100%)*	*(N = 87; 47.3%)*	*(N = 97; 52.7%)*	
**Surgical approach, *n* (%)**				0.560
**Open**	142 (77)	70 (80)	72 (74)	
**Laparoscopic**	11 (6)	5 (6)	6 (6)	
**Robotic**	31 (17)	12 (14)	19 (20)	
**Associated urethrectomy, *n* (%)**				0.239
**Yes**	167 (91)	81 (93)	86 (89)	
**No**	16 (8)	5 (6)	11 (11))	
**NA**	1 (1)	1 (1)	0 (0)	
**Lymphadenectomy, *n* (%)**				0.594
**No**	7 (4)	4 (5)	3 (3)	
**Yes**	177 (96)	83 (95)	94 (97)	
**Urinary diversion, *n* (%)**				0.930
**Ileal conduit**	94 (51)	46 (53)	48 (50)	
**Enterocystoplasty**	80 (44)	37 (43)	43 (44)	
**Skin ureterostomies**	10 (5)	4 (4)	6 (6)	
**Median operative time, (IQR)**	300 (258–390)	315 (250–420)	300 (260–390)	0.948
**Median blood loss, (IQR)**	800 (400–1000)	800 (400–1000)	700 (400–1000)	0.498
**Blood transfusion, *n* (%)**				0.882
**No**	105 (57)	48 (55)	57 (59)	
**Yes**	43 (23)	21 (24)	22 (23)	
**NA**	36 (20)	18 (21)	18 (18)	

**Table 3 cancers-14-03797-t003:** Postoperative outcomes of included patients treated with upfront or delayed eRC for HR-NMIBC.

Postoperative Outcomes	Total	Upfront eRC	Delayed eRC	*p*
	*(N = 184; 100%)*	*(N = 87; 47.3%)*	*(N = 97; 52.7%)*	
**Median length of stay, days (IQR)**	17 (14–21)	17 (14–21)	17 (14–21)	0.333
**Blood transfusion, *n* (%)**				0.625
**No**	115 (62)	52 (60)	63 (65)	
**Yes**	42 (23)	20 (23)	22 (23)	
**No**	27 (15)	15 (17)	12 (12)	
**Median hemoglobin level at discharge, g/dL (IQR)**	11 (10–13)	10 (9–11)	11 (10–12)	0.612
**Median creatinine level at discharge, ng/mL (IQR)**	93 (74–113)	96 (79–114)	88 (72–112)	0.132
**Complications (according to Clavien–Dindo scale), *n* (%)**				0.501
**No**	73 (40)	38 (44)	35 (36)	
**Yes**				
**1**	17 (9)	9 (10)	8 (8)	
**2**	48 (26)	21 (24)	27 (28)	
**3**	34 (18)	12 (14)	22 (23)	
**4**	8 (5)	4 (5)	4 (4)	
**5**	4 (2)	3 (3)	1 (1)	

**Table 4 cancers-14-03797-t004:** Cross-tabulation of cT stage from TURBT and pT stage from eRC of included patients with HR-NMIBC.

cT Stage (on TURBT Specimen), *n* (%)	pT Stage (on eRC Specimen), *n* (%)						Total
	pT0N0-x	pTaN0-x	pTisN0-x	pT1N0-x	pT2N0-x	≥pT3 and/or pN+	
**cTa**	7 (21)	12 (67)	10 (24)	4 (11)	4 (22)	1 (1)	38 (100)
**cTis**	3 (9)	0 (0)	13 (32)	3 (8)	1 (6)	3 (9)	23 (100)
**cT1**	24 (70)	6 (34)	18 (44)	29 (81)	13 (72)	33 (90)	123 (100)
**Total**	34 (19)	18 (10)	41 (22)	36 (19)	18 (10)	37 (20)	184 (100)

**Table 5 cancers-14-03797-t005:** Pathological outcomes of included patients treated with upfront or delayed eRC for HR-NMIBC. Bold: It is a significant result.

Pathological Outcomes	Total	Upfront eRC	Delayed eRC	*p*
	*(N = 184; 100%)*	*(N = 87; 47.3%)*	*(N = 97; 52.7%)*	
**Histology, *n* (%)**				**0.027**
**Pure urothelial**	144 (78)	75 (86)	69 (71)	
**variant**	6 (3)	3 (4)	3 (3)	
**No residual tumor**	34 (19)	9 (10)	25 (26)	
**pT stage, *n* (%)**				**0.046**
**pT0**	34 (19)	9 (10)	25 (26)	
**pTa**	19 (10)	10 (12)	9 (9)	
**pTis**	41 (22)	16 (18)	25 (26)	
**pT1**	37 (20)	22 (25)	15 (16)	
**pT2**	21 (12)	12 (14)	9 (9)	
**≥pT3**	32 (17)	18 (21)	14 (14)	
**pN stage, *n* (%)**				0.757
**pN0**	164 (89)	76 (87)	88 (91)	
**pN+**	13 (7)	7 (8)	6 (6)	
**pNx**	7 (4)	4 (5)	3 (3)	
**Concomitant CIS, *n* (%)**				0.613
**Absent**	103 (56)	47 (54)	56 (58)	
**Present**	81 (44)	40 (46)	41 (42)	
**LVI, *n* (%)**				0.057
**Absent**	115 (63)	58 (67)	57 (59)	
**Present**	6 (3)	5 (6)	1 (1)	
**NA**	63 (34)	24 (27)	39 (40)	
**Surgical margin, *n* (%)**				0.260
**Negatives**	174 (95)	84 (97)	90 (93)	
**Positives**	10 (5)	3 (3)	7 (7)	

**Table 6 cancers-14-03797-t006:** Uni- and multivariate analysis of predictors of pathological upstaging in included patients treated with upfront or delayed eRC for HR-NMIBC. Bold: It is a significant result.

Predictive Factors	Univariate Analysis		Multivariate Analysis	
	Odds Ratio (IC 95%)	*p*	Odds Ratio (IC 95%)	*p*
**eRC**	Ref		Ref	
**Delayed**	1.68 (0.89–3.18)	0.109	2.65 (1.23–5.67)	**0.012**
**Upfront**				
**Age**	1.01 (0.98–1.04)	0.428	-	-
**Gender**	Ref			
**Male**	1.52 (0.31–7.56)	0.510	-	-
**Female**				
**BMI**	1.92 (1.86–1.99)	**0.025**	1.29 (1.22–1.97)	**0.008**
**ASA score, *n* (%)**	Ref			
**≤2**	1.42 (0.71–2.84)	0.990	-	-
**≥3**	1.04 (0.19–5.61)	0.050		
**NA**				
**Smoking status, *n* (%)**	Ref		Ref	
**Never**	3.55 (1.15–10.9)	**0.028**	5.40 (1.45–20.1)	**0.012**
**Weaned**	1.94 (0.43–2.06)	0.881	1.72 (0.30–1.71)	0.461
**Active**	0.52 (0.05–4.89)	0.565	0.60 (0.05–7.73)	0.696
**NA**				
**cT stage, *n* (%)**	Ref			
**cTa**				-
**cTis**	1.21 (0.38–3.88)	0.743	-	
**cT1**	1.02 (0.49–2.14)	0.950		
**Focality, *n* (%)**	Ref		Ref	
**Unifocal**	1.94 (0.63–3.45)	0.129	1.89 (0.74–4.79)	0.181
**Multifocal**	3.63 (1.45–9.09)	0.006	6.01 (2.07–17.5)	**0.001**
**NA**				
**Size, *n* (%)**	Ref			
**<3 cm**	1.01 (0.43–2.39)	0.982	-	-
**≥3 cm**	0.67 (0.31–1.45)	0.308		
**NA**				
**Concomitant CIS, *n* (%)**	Ref		Ref	
**Absent**	1.57 (1.30–2.08)	0.084	1.46 (1.22–1.95)	**0.037**
**Present**				
**LVI, *n* (%)**	Ref			
**Absent**	2.89 (0.77–10.7)	0.115	-	-
**Present**	1.41 (0.73–2.74)	0.306		
**NA**				
**Second look TURB, *n* (%)**	Ref			
**No**	0.66 (0.26–1.71)	0.398	-	-
**Yes**	0.54 (0.09–3.37)	0.511		
**NA**				

**Table 7 cancers-14-03797-t007:** Propensity score model that predicted the receipt of upfront vs. delayed eRC in included patients with HR-NMIBC. Bold: It is a significant result.

Predictive Factors	Odds Ratio (IC 95%)	*p*
**Age**	1.03 (1.01–1.07)	**0.039**
**Gender**	Ref	
**Male**	0.71 (0.13–3.84)	0.695
**Female**		
**BMI**	1.01 (0.94–1.08)	0.832
**ASA score, *n* (%)**	Ref	
**≤2**	0.49 (0.21–1.11)	0.088
**≥3**	4.69 (0.41–53.9)	0.215
**NA**		
**Smoking status, *n* (%)**	Ref	
**Never**	0.79 (0.19–3.30)	0.752
**Former**	1.04 (0.45–2.40)	0.934
**Current**	1.44 (0.18–11.6)	0.730
**NA**		
**cT stage, *n* (%)**	Ref	
**cTa**		
**cTis**	0.32 (0.07–1.54)	0.156
**cT1**	0.97 (0.45–2.10)	0.935
**Tumor focality, *n* (%)**	Ref	
**Unifocal**	1.47 (0.63–3.45)	0.372
**Multifocal**	0.81 (0.30–2.23)	0.686
**NA**		
**Tumor size, *n* (%)**	Ref	
**<3 cm**	3.13 (1.00–9.79)	0.050
**≥3 cm**	1.06 (0.43–2.62)	0.894
**NA**		
**Concomitant CIS, *n* (%)**	Ref	
**Absent**	0.87 (0.42–1.80)	0.706
**Present**		
**LVI, *n* (%)**	Ref	
**Absent**	0.49 (0.08–3.06)	0.443
**Present**	0.70 (0.31–1.58)	0.386
**NA**		
**Second look TURB, *n* (%)**	Ref	
**No**	0.06 (0.01–0.35)	**0.001**
**Yes**	0.02 (0.01–0.33)	**0.006**
**NA**		

## Data Availability

Data available on request due to restrictions, e.g., privacy or ethical.

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
