# Peer review of "Effectiveness of Early Radical Cystectomy for High-Risk Non-Muscle Invasive Bladder Cancer"

_cancers, 2022, doi:10.3390/cancers14153797_

Round 1
Reviewer 1 Report
The authors of the manuscript entitled “Effectiveness of Early Radical Cystectomy for High-Risk Non-2 Muscle Invasive Bladder Cancer” should be congratulated on the interesting work presented.
The authors reported postoperative and oncological outcomes of patients that undergo radical cystectomy after a diagnosis of NMIBC; The sample is stratified according to whether the patient received BCG before cystectomy or underwent radical cystectomy as first-line treatment.
The topic is certainly interesting; it is essential to identify patients who could benefit from an initial conservative approach without leading to worse outcomes.
The manuscript is clear, but some imperfections should be resolved for publication.
First it is not described the previous treatment of the patients that underwent BCG instillations.
On line 176, the authors report that these patients received a median of 6 instillations, the authors should specify what the numbers in brackets refer to (IQR?). The authors should also specify how many patients received adequate treatment with BCG and what schedule was followed for the eventual maintenance instillations. It seems that most of these patients underwent only BCG induction (do they recur after induction?), but the median time to RC is more than one year; please explain this.
Did any patients undergo other intravesical agents after BCG failure and before RC?
According to the FDA definition, how many patients were truly BCG Unresponsive?
Even if it appears obvious, the authors should state in methods what they intend as “early cystectomy” (any Cystectomy in NMBIC patients?).
In my opinion, figures 4 and 5 could be uploaded as supplementary material.
Line 46, please correct the word “blader”
Line 297, “there was,” is repeated twice.
Author Response
Point 1: First it is not described the previous treatment of the patients that underwent BCG instillations.
On line 176, the authors report that these patients received a median of 6 instillations, the authors should specify what the numbers in brackets refer to (IQR?). The authors should also specify how many patients received adequate treatment with BCG and what schedule was followed for the eventual maintenance instillations. It seems that most of these patients underwent only BCG induction (do they recur after induction?), but the median time to RC is more than one year; please explain this.
Did any patients undergo other intravesical agents after BCG failure and before RC?
According to the FDA definition, how many patients were truly BCG Unresponsive?
Response 1:
Dear reveiwer, thank you for your careful review and for all your comments and questions.
- In our study, the group of patients who underwent upfront eRC had received at least an induction course of BCG intravesical instillation. Unfortunately, because of the retrospective and multicentric nature of our study, it was not possible to obtain details about the number of instillations recevied during the maintenance courses. We are aware that this information is relevant to the interpretation of the data, and the presentation of this limitation was unclear in our manuscript. We modified it in the material and methods part (line 96-97) and we removed the results (line 177-178). Thus, a clear definition of patients truly BCG unresponsive, according to FDA, was not possible in our study.
- To our knowledge, no patients underwent other intravesical agents than BCG after BCG failure
Point 2 : Even if it appears obvious, the authors should state in methods what they intend as “early cystectomy” (any Cystectomy in NMBIC patients?)
Response 2 : Thanks for your feedback, we have amended the manuscript accordingly (line 62-63)
Reviewer 2 Report
In the study” Effectiveness of Early Radical Cystectomy for High-Risk Non- Muscle Invasive Bladder Cancer” Diamant and co-authors address an important issue in the management of high-risk NMIBC. The data presented argues the eRC confers no survival benefit. This information could be of value to surgeons and oncologist in decision making. As the authors admit, the results may differ if there is additional stratification of patients, but that would require a much larger patient cohort.
One troubling aspect of the study is the observed pathological upstaging evident in eRC. The authors note that it is conceivable that patients selected for eRC presented with more aggressive pathological features at initial diagnosis than those treated with intravesical instillations of BCG. This is important data to have since delaying RC for tumors with more aggressive pathological features could be deleterious to survival. The upstaging was not detected when RC was delayed, and the patients underwent BCG treatment. Is there any evidence BCG treatment itself alters pathological features? Could the authors comment on this if they have additional insight?
Given these results the decision making continues to have little evidence-based guidance. If delay truly does not affect survival but enhances patient quality of life that should be a major factor in decision making. However, given the upstaging, the decision may not be that simple.
Author Response
Point 1: . In the study” Effectiveness of Early Radical Cystectomy for High-Risk Non- Muscle Invasive Bladder Cancer” Diamant and co-authors address an important issue in the management of high-risk NMIBC. The data presented argues the eRC confers no survival benefit. This information could be of value to surgeons and oncologist in decision making. As the authors admit, the results may differ if there is additional stratification of patients, but that would require a much larger patient cohort.
One troubling aspect of the study is the observed pathological upstaging evident in eRC. The authors note that it is conceivable that patients selected for eRC presented with more aggressive pathological features at initial diagnosis than those treated with intravesical instillations of BCG. This is important data to have since delaying RC for tumors with more aggressive pathological features could be deleterious to survival. The upstaging was not detected when RC was delayed, and the patients underwent BCG treatment. Is there any evidence BCG treatment itself alters pathological features? Could the authors comment on this if they have additional insight?
Given these results the decision making continues to have little evidence-based guidance. If delay truly does not affect survival but enhances patient quality of life that should be a major factor in decision making. However, given the upstaging, the decision may not be that simple.
Response 1: Thank you for your comments. Indeed, in our study we found a higher risk of upstaging for patients who underwent upfront eRC. Our hypothesis was first a selection bias. Actually, the proposition of eRC was made to patients who presented poor prognostic factors, including LVI, histologic variants, multifocality, CIS, large tumor size, pT1b, or combination of several of factors previously mentioned. A significant number of them presented actually a MIBC, which was not diagnosed with our classic tools. Another explanation, more hypothetical, was the role of BCG instillation. We assumed that BCG may contribute, through its immunological mechanism, to the limitation, at least momentarily, of local invasion, and thus delay the time of eRC, despite the high-risk character of their pathology. No robust data regarding this hypothesis were found in the literature. In addition, our study allows us to hypothesize that there would be no loss of chance for patients initially treated with BCG therapy followed by a delayed eRC, but the benefit of a strategy based on pre-BCG therapy might even be more important. Indeed, patients initially treated with BCG therapy and never receiving delayed eRC in the absence of recurrence were not considered for inclusion in our study. It is conceivable that those patients have a favorable prognosis and if they had been included in the comparator arm, the impact of upfront eRC on oncologic outcomes would have been more pejorative, providing further incentive to propose BCG therapy first for the management of HRNMIBC.